# CurQ+, a Next-Generation Formulation of Curcumin, Ameliorates Growth Plate Chondrocyte Stress and Increases Limb Growth in a Mouse Model of Pseudoachondroplasia

**DOI:** 10.3390/ijms24043845

**Published:** 2023-02-14

**Authors:** Jacqueline T. Hecht, Alka C. Veerisetty, Mohammad G. Hossain, Frankie Chiu, Karen L. Posey

**Affiliations:** 1Department of Pediatrics, McGovern Medical School at UTHealth Houston, Houston, TX 77030, USA; 2School of Dentistry, The University of Texas Health Science Center at Houston (UTHealth), Houston, TX 77030, USA

**Keywords:** cartilage oligomeric matrix protein, COMP, autophagy, curcumin, dwarfism, chondrocyte, articular cartilage, joint degeneration

## Abstract

Mutations in cartilage oligomeric matrix protein (COMP) causes protein misfolding and accumulation in chondrocytes that compromises skeletal growth and joint health in pseudoachondroplasia (PSACH), a severe dwarfing condition. Using the MT-COMP mice, a murine model of PSACH, we showed that pathological autophagy blockage was key to the intracellular accumulation of mutant-COMP. Autophagy is blocked by elevated mTORC1 signaling, preventing ER clearance and ensuring chondrocyte death. We demonstrated that resveratrol reduces the growth plate pathology by relieving the autophagy blockage allowing the ER clearance of mutant-COMP, which partially rescues limb length. To expand potential PSACH treatment options, CurQ+, a uniquely absorbable formulation of curcumin, was tested in MT-COMP mice at doses of 82.3 (1X) and 164.6 mg/kg (2X). CurQ+ treatment of MT-COMP mice from 1 to 4 weeks postnatally decreased mutant COMP intracellular retention, inflammation, restoring both autophagy and chondrocyte proliferation. CurQ+ reduction of cellular stress in growth plate chondrocytes dramatically reduced chondrocyte death, normalized femur length at 2X 164.6 mg/kg and recovered 60% of lost limb growth at 1X 82.3 mg/kg. These results indicate that CurQ+ is a potential therapy for COMPopathy-associated lost limb growth, joint degeneration, and other conditions involving persistent inflammation, oxidative stress, and a block of autophagy.

## 1. Introduction

Cartilage oligomeric matrix protein (COMP) is a non-collagenous extracellular matrix (ECM) glycoprotein that assists in collagen fibril assembly and binds to multiple ECM proteins enhancing cartilage integrity [1,2,3,4,5,6,7]. Mutations in COMP result in protein misfolding causing two well-characterized skeletal dysplasias: PSACH and multiple epiphyseal dysplasia (MED) [8]. The clinical features of PSACH include disproportionate short stature, short fingers and toes, joint laxity, attractive angular face, and a waddling gait [8,9]. The dramatic short stature of PSACH results in an adult height that approximates that of an average stature 6-year-old [8,10,11,12]. Severe life-long joint pain and precocious joint degeneration are the most debilitating complications. Currently, treatments are only short-term pain relievers and, ultimately, replacement of various joints, since targeted medical management strategies are not available, [8,11].

In 1973, PSACH was classified as an ER storage disorder, because of the massive retention of a lamellar appearing material in the ER of PSACH growth plate chondrocytes [1,13,14,15,16]. Our subsequent studies showed that COMP and other ECM proteins that interact with COMP were retained within chondrocytes and this material in the ER-compromised cellular function, causing premature chondrocyte death and ultimately short stature [12,14,17,18,19,20,21]. The mutant (MT)-COMP mouse expressing mutant-D469del COMP protein was generated to study the underlying pathologic mechanisms and replicates both clinical and chondrocyte characteristics of PSACH [22,23,24,25,26]. Doxycycline (DOX) expression of mutant-COMP protein stimulates persistent ER stress leading to both inflammation and oxidative stress that in turn exacerbates ER stress, thereby generating a self-perpetuating pathological loop between ER stress, inflammation, and oxidative stress processes [25,27,28,29,30,31]. Specifically, we have shown that mutant-COMP accumulation generates multiple cellular stresses in the MT-COMP growth plate chondrocytes, including ER stress (CHOP), oxidative stress, inflammation (TNFα), ECM degradative enzymes (MMP-13), block of autophagy (pS6; LC3 positive vesicles) [18,32]. ER stress and tumor necrosis factor alpha (TNFα) together increased midline 1 (MID1), a direct stimulator of mTORC1 signaling [18,26,32]. High levels of mTORC1 signaling favor protein expression over macroautophagy (hereafter referred to as autophagy). Autophagy blockage drives continuous accumulation of mutant-COMP and other ECM proteins in the ER of chondrocytes, building an intracellular matrix that is not degraded by the cellular machinery, eventually overwhelming the cellular stress coping mechanisms and compromising viability [18,26,32]. Chondrocyte death is the result of oxidative stress that induces DNA damage and, ultimately, caspase-independent necroptosis [19,33]. Importantly, these stresses were not observed in MT-COMP mice in the absence of DOX.

We have previously demonstrated that decreasing chondrocyte stress (inflammation, oxidative stress, and block of autophagy) in MT-COMP mice with resveratrol treatment sustained chondrocyte function allowing a partial rescue of limb growth and preserving joint health [18,34]. Similarly, curcumin targets a number of processes that are involved in the mutant-COMP protein pathology including inflammation, oxidative stress, and autophagy [35,36,37,38,39], indicating that curcumin may be therapeutic for PSACH. Moreover, our previous treatment of MT-COMP mice with turmeric (curcumin, an active ingredient of turmeric) suggested that purified curcumin might reduce the accumulation of mutant-COMP [24]. CurQ+ is a unique coconut oil-based dispersion of curcumin that dramatically increases absorption (up to 65-fold) over powdered curcumin [40]. This study assessed whether CurQ+ dampened the MT-COMP growth plate pathology. 

## 2. Results

### 2.1. CurQ+ Reduced Intracellular Retention of Mutant-COMP in Growth Plate Chondrocytes

The most distinctive feature of PSACH chondrocytes is the intracellular retention of misfolded mutant-COMP in the ER [15] in both growth plates and articular chondrocytes [25,26]. Administration of CurQ+ from post-natal day (P) P7-P28 dramatically reduced intracellular retention of mutant-COMP protein in growth plate chondrocytes compared to untreated MT-COMP mice (Figure 1). CurQ+ treatment of MT-COMP growth plates at 2X× dose had a greater impact with apparent limiting of intracellular mutant-COMP compared to 1X dose (Figure 1C–F). Quantification of intracellular COMP using Image J showed that treatment with CurQ+ reduced intracellular COMP to approximately 4% of untreated MT-COMP controls (Appendix A). Notably, mutant human COMP protein was exported to the ECM in the growth plate resting zone (Figure 1C,E). 

### 2.2. CurQ+ Treatment Decreased Markers of Inflammation and Normalized Proliferation and Autophagy in the MT-COMP Growth Plate

Growth plate chondrocytes are known to produce proinflammatory cytokines [41]; we have shown that mutant-COMP expression increased expression of multiple inflammatory proteins [18,20,24,26,32]. Figure 2 shows that CurQ+ treatment from P7–P28 repressed IL6 inflammation, restored autophagy, and reestablished chondrocyte proliferation in MT-COMP growth plates compared to the untreated controls. IL6, a proinflammatory cytokine increased by mutant-COMP protein expression [18,20,24,26,32], was dampened by CurQ+ treatments (Figure 2A–D), with 2X dose having a greater impact. 

Misfolded proteins in the ER are eliminated by two major pathways: autophagy or proteasomal degradation [42]. To define which pathway is responsible for mutant-COMP accumulation, inhibitors for both processes (autophagy, proteasomal degradation) were tested in combination with resveratrol. Experimental inhibition of autophagy and increased number of LC3-II positive vesicles (autophagic vesicles) indicated that autophagy was responsible for resveratrol driven mutant-COMP clearance [32]. pS6 is a readout for mTORC1 signaling [43] that controls autophagy. When pS6 levels are low (mTORC1 signaling), autophagy is stimulated, while in contrast, high levels of pS6 (mTORC1 signaling) protein synthesis and survival is favored and autophagy is repressed [44,45]. As shown in Figure 2I–L, pS6 was decreased indicating de-repression of autophagy, which was again higher in the 2X CurQ+ treated growth plate.

Chondrocyte proliferation and death, important components of growth plate homeostasis, are markedly abnormal in MT-COMP growth plate chondrocytes (Figure 2B,F,N,R). CurQ+ treatment restored chondrocyte proliferation (Figure 2Q–T) and substantially dampens cell death as evidenced by reduced TUNEL staining (Figure 2M–P). The number of TUNEL-positive chondrocytes in the MT-COMP treated mice with CurQ+ (O, P) were restored to baseline levels (M), while untreated MT-COMP mice had numerous TUNEL-positive chondrocytes throughout the growth plate (Figure 2N). CurQ+ 2X treatment improved PCNA signal more than 1X while chondrocyte death (TUNEL) was greatly diminished at both dosages. Curcumin reportedly reduces osteoarthritis (OA) joint degeneration [46] by stimulating autophagy [36,37,47,48] and/or increasing the expression of IL10 [35,46,49], which reduces matrix metalloproteases (MMPs) that degrade the ECM of cartilage [46]. As shown in Figure 2E–H, CurQ+ treatment restored IL10 expression, suggesting increased anti-inflammatory action, especially with the 2X dose. Chondrocyte proliferation and death, important components of growth plate homeostasis, are markedly abnormal in MT-COMP growth plate chondrocytes. Retention of mutant COMP protein in ER of growth plate chondrocytes is associated with widespread cell death in PSACH and MT-COMP mice [13,14,25,50,51,52,53]. Normalization of growth plate chondrocyte survival would be expected to support restored growth plate function in MT-COMP mice. 

### 2.3. Further Evidence That CurQ+ Treatment Decreases Inflammation, ER Stress, and Increases Anti-Inflammatory Activity

Since 2X CurQ+ generated a better outcome in the growth plate, we assessed additional markers for inflammation (TNFα; IL1β), ER stress (CHOP), autophagy block (MID1), and cartilage degeneration (MMP13) in 2X CurQ+ treated mice. Previously, we showed that both TNFα and IL1β are elevated in MT-COMP growth plates [18,20,24,26,32] and are important inflammation drivers in the mutant-COMP protein pathology. Figure 3 shows that both TNFα and IL1β are elevated in MT-COMP growth plates (Figure 3B,E) and 2X CurQ+ treatment decreased inflammation (Figure 3C,F) to control levels (Figure 3A,D). This outcome supports the previously reported anti-inflammatory action of curcumin [35,46,49,54]. 

Previously, we demonstrated that all three ER stress sensors (PERK, ATF6, IRE1-XBP1) were activated by mutant-COMP expression early in the pathologic process but only the PERK pathway proceeded beyond activation leading to increased CHOP expression [19,33]. Evaluation of Chop, Xbp1, Gadd45a, Gadd34, Ero1b, Grp78, and ER degradation enhancing alpha-mannosidase-like protein 1 mRNA levels of (Edem1) [19] showed that Edem1 mRNA level was not changed in MT-COMP mice suggesting that endoplasmic-reticulum-associated protein degradation (ERAD) did not play a substantial role in the mutant-COMP pathology. Importantly, we demonstrated that ER stress (CHOP), fundamental to the mutant-COMP protein pathology, is dramatically reduced by CurQ+ treatment (Figure 3G–I), thereby decreasing MID1 levels (Figure 3J–L), which dampens a process that blocks autophagy preventing mutant-COMP protein clearance. IL10, an anti-inflammatory protein, has been shown to reduce MMP13, an enzyme that degrades cartilage [46]. CurQ+ 2X treatment increased IL10 levels (Figure 2H) and consistent with its known action, MMP13 levels were reduced (Figure 3M–O).

### 2.4. MT-COMP Lost Femoral Length Was Rescued by CurQ+ Treatment

Previously, we showed that resveratrol and aspirin rescued approximately 50% of the lost limb growth in untreated MT-COMP mice and dampened the mutant-COMP cellular stress in growth plate chondrocytes [24]. Figure 4 shows that CurQ+ treatment restored 60% of lost limb growth at 1X and fully restored limb length at 2X dose.

## 3. Discussion

The results of this study show that the CurQ+, curcumin with a unique coconut oil-based dispersion technology, resolved the growth plate chondrocyte stress in the MT-COMP murine model of PSACH. Importantly, this treatment restored lost femoral length. This dramatic result occurred after three weeks of CurQ+ treatment, which reduced intracellular accumulation of COMP, inflammation, and chondrocyte death, and increased autophagy, IL10 expression, and proliferation throughout the MT-COMP growth plate. These are important measures of therapeutic efficacy since (1) inflammation is linked to PSACH joint pain, (2) autophagy blockage prevents mutant-COMP clearance from chondrocytes and jeopardizes chondrocyte viability and, (3) chondrocyte proliferation is crucial to growth plate function.

Preservation of chondrocyte function and longevity are essential to endochondral linear bone growth as chondrocytes generate the cartilage template that calcifies laying down new bone resulting in bone growth. Importantly, 2X CurQ+ treatment restored normal femur length and 1X CurQ+ significantly increased femur length compared to the untreated MT-COMP femurs (Figure 3). Unexpectedly, CurQ+ treatment allowed some export of mutant-COMP from resting zone growth plate chondrocytes suggesting that some mutant-COMP protein was able to escape the ER quality control system and be exported. CurQ+ was selected based on a long history of safe human consumption of curcumin/turmeric, and because curcumin has been shown to reduce joint degeneration in OA [37,49,55,56] and cellular stresses that are involved in the mutant-COMP pathology [18,20,24,26,32]. Moreover, CurQ+, a formulation of curcumin, was specifically evaluated because it is more absorbable compared to 95% powdered curcumin [40]. 

The current working model of PSACH growth plate chondrocyte pathology is a self-perpetuating loop initiated by the retention of mutant-COMP in the ER of chondrocytes that stimulates severe and unrelenting cellular stress resulting in chondrocyte death. The loss of growth plate chondrocytes decreases the pool of chondrocytes that produce matrix for long bone growth and the dead/dying chondrocytes may release inflammatory molecules that compromise viability and function of the remaining chondrocytes. This mechanistic pathway has been well-described in growth plate chondrocytes in our MT-COMP mouse that models the PSACH clinical and chondrocyte pathology. Intracellular accumulation of mutant-COMP in the ER of growth plate chondrocytes is the initiating step in the MT-COMP stress pathology [10,11,15] and CurQ+ not only relieves the mutant-COMP protein accumulation but also allows some of the protein to be exported to the extracellular matrix (Figure 1). Our recent work shows that autophagy blockage is the key pathological process that prevents mutant-COMP from being cleared from the ER of growth plate chondrocytes [18,26,32]. However, many of the stress processes likely interact with each other further stimulating cellular stress (autophagy block, ER stress, and oxidative stress) [18,19,20,24,26,33,57]. This interconnection of pathological stresses suggests that reducing one stress should lead to a reduction in other stress processes; this was observed with CurQ+ and, previously, with resveratrol [18,26,32]. This crosstalk complicates identifying the direct action of CurQ+. 

The most consistently reported curcumin benefits are inhibition of (1) mTORC1 resulting in autophagy stimulation [36,37,47,48], (2) antioxidant processes [58], and (3) anti-inflammatory activity. In the MT-COMP mouse, CurQ+ treatment decreased IL6, IL1β, and TNFα inflammation, autophagy blockage, chondrocyte death, and increased proliferation and IL10 (anti-inflammatory). Additionally, the anti-inflammatory effect of CurQ+ that is mediated through multiple cytokines suggests that CurQ+ may be an effective method to address pain in PSACH that has been associated with inflammation [8,10,59,60]. Importantly, CurQ+ preserved growth plate chondrocyte viability and proliferation. CurQ+ restored IL10 levels, which control MMP13 cartilage degradation and decreased pro-inflammatory molecules IL6, IL1β, and TNFα that drive high levels of chondrocyte stress. IL1β and TNFα attract eosinophils, which we previously showed were part of the mutant-COMP protein pathology [61]. CurQ+ stimulation of autophagy and anti-inflammatory activity supports chondrocyte survival while IL10 ultimately reduces cartilage degradation suggesting that these are the essential mechanisms of action of CurQ+ in MT-COMP mice. Future testing in clinical trials would ideally target these pathologic processes early in life to prevent childhood joint pain and potentially restore some lost limb growth.

Previously, we have shown that turmeric treatments permit a small amount of extracellular export of mutant-COMP [24]; curcumin is the active ingredient in turmeric. Interestingly, with CurQ+ treatment, mutant-COMP was observed in the extracellular matrix surrounding the resting zone chondrocytes, suggesting that ER quality control mechanisms that retain misfolded protein were either bypassed or overridden. CurQ+ stimulated export was limited to the resting zone and was not observed in the proliferating and hypertrophic zone of the growth plate at 4 weeks. Our previous resveratrol work demonstrated that autophagy clears mutant-COMP protein [18,20,24,26,32], and this is likely the mechanism by which mutant-COMP protein is eliminated from the proliferating and hypertrophic chondrocytes with CurQ+ treatment. Curcumin has been shown to induce autophagy protecting cellular viability [62] and decreases pS6 (mTORC1 signaling marker), indicating that curcumin influences autophagy. Interestingly, 1X CurQ+-treated resting zone growth plate chondrocytes had more mutant-COMP protein in the extracellular matrix than the 2X dose. Autophagy levels are perhaps higher in 2X CurQ+, which reduces mutant-COMP export. It is not known how some mutant-COMP escapes ER quality control systems and autophagy, but lower levels of protein synthesis in the resting zone may not strongly trigger ER clearance mechanisms allowing the secretion of mutant-COMP. Since professional secretory cells, such as chondrocytes, have basal ER stress that is essential for differentiation [63], resting zone differentiation ER stress may allow export, while not in chondrocytes in other zones. Alternatively, there could be different secretory factors or ER quality control mechanisms in the resting zone that supports export that is not present in other growth plate zones. This is supported by the observation that there is differential expression of ER stress-related mRNAs driven by ER stress in the zones of the growth plate in another dwarfing condition, Schmid metaphyseal chondrodysplasia (MCDS) [64].

It is not known how the presence of mutant-COMP may impact the ECM of the growth plate cartilage, but given the many COMP binding partners that may be affected by the presence of misfolded COMP, it is likely that the ECM quality would be negatively altered. The effect of mutant-COMP protein on the matrix will be evaluated in long-term CurQ+ treatment studies (20 wks) that are currently underway and designed to measure the impact of CurQ+ on skeletal growth into adulthood and on joint health. If mutant-COMP protein in the matrix negatively impacts growth plate function, we expect that growth of treated mice would not keep pace with controls although our current work shows CurQ+ prevented loss of limb growth during the first 4 weeks of mouse life. However, if CurQ+ prevents articular chondrocyte dysfunction and mutant-COMP protein is not exported to the ECM of the articular cartilage, we anticipate that joint degeneration would be inhibited.

The most important outcome of this study is the improvement in long bone growth, which results from CurQ+ reduction in MT-COMP growth plate chondrocyte stress. The 2X CurQ+ dosage fully restores femur length while 1X dosage recovers 60% of lost femoral length. Importantly, at these dosages, weight and size were normal and did not compromise pup health. Moreover, numerous reports have demonstrated that curcumin has chondroprotective effects that slow/inhibit joint degeneration [35,36,37,49,54,65,66,67,68,69], a common complication in PSACH. Therefore, CurQ+ should also be considered for PSACH therapy to target prevention/halting of joint degeneration. Future studies in MT-COMP mice will focus on joint degeneration, since PSACH pathology is not limited to the growth plate; premature joint degeneration is the most debilitating and painful complication associated with this COMPopathy [8,11,17,21,34,50,70]. All joints degenerate beginning in the teen years and joint replacements are the only treatment (replacements beginning at 25–30 years). Joint replacement therapy is not ideal for PSACH, since not all joints are replaceable, and the limited life span of implanted joints means that multiple replacements of the same joint are required over a lifetime. A joint-sparing, nonsurgical therapy would make a profound difference in the quality of life in PSACH. Given the reduction in the mutant-COMP stress in the growth plate chondrocytes, and restoration of limb growth, CurQ+ should be tested for its ability to preserve joint health in the MT-COMP mouse, working towards the goal of a comprehensive nonsurgical therapy for PSACH.

## 4. Materials and Methods

### 4.1. Bigenic Mice

MT-COMP mice were generated by introducing DNA-containing expression cassettes derived from two plasmids, pTRE-MT-COMP (D469del-COMP mutation) and pTET-On-Col II, as previously described [25]. Mice expressed D469del-COMP protein in all tissues expressing type II collagen, in the presence of doxycycline (DOX 500 ng/mL), administered through drinking water (with 5% wt/vol sucrose) pre- and postnatally. All mutant mice were healthy and reproduced. C57BL/6 mice were used as controls, since the wild-type (WT)-COMP mice showed no phenotypic differences in our previous studies [25]. Mice were housed in single-sex groups after weaning (P21) and fed standard chow (PicoLab rodent diet 20 #5053, LabDiet, St. Louis, MO, USA). The Animal Welfare Committee at the University of Texas Medical School at Houston approved these studies and all experiments complied with the Guide for the Care and Use of Laboratory Animals: Eighth Edition, ISBN-10: 0-309-15396-4 and NIH guidelines.

### 4.2. Drug Administration

DOX (500 ng/mL) was administered through drinking water with 5% wt/vol sucrose to increase palatability. CurQ+ was administered by gavage 5 days per week beginning at P7 until joint collection at P28. The dose was 82.3 mg/kg (human equivalent dose assuming a 60 kg human) for 1X and 164.6 mg/kg for 2X doses. The 1X dose was selected based on FDA Guidance on Human Equivalent Dose Determination and Stratum Nutrition (Carthage, MO, USA) recommendations based on their work with rats [49]. CurQ+ was provided by Stratum Nutrition. 

### 4.3. Immunohistochemistry

Hind limbs MT-COMP and C57BL/6 control mice (both sexes) were collected at P28, and tibial growth plates were analyzed as previously described [25]. Mouse limbs were fixed in 95% vol/vol ethanol for immunostaining for human COMP (Abcam Cambridge, MA ab11056-rat 1:100), rabbit polyclonal COMP 1:300 (Kamiya, Seattle, WA, USA), goat polyclonal Type II collagen 1:200 (Santa Cruz Biotechnology, Dallas, TX, USA), pS6 (1:200 2215S rabbit polyclonal Cell Signaling Technology, Danvers, MA, USA), interleukin 16 (IL16) (Santa Cruz Biotechnology; sc-7902, 1:100), PCNA staining kit (Invitrogen, Frederick, MD 93-1143), IL10 (1:200 bs-0698R Bioss Antibodies Woburn, MA, USA) or in 10% wt/vol formalin for terminal deoxynucleotidyl transferase-mediated deoxyuridine triphosphate-biotin nick-end labeling (TUNEL) staining. The COMP rat antibody does not cross-react with endogenous mouse COMP and only recognizes human COMP. Species-specific biotinylated secondary antibodies were incubated with each section for 1 h following streptavidin/HRP incubation and chromogen detection for visualization. At least 8 C57BL/6 and 8 MT-COMP mice were examined for each treatment.

### 4.4. Limb Length Measurements

Hind limbs were obtained from 10 male mice for each group and the soft tissue was carefully removed. The hind limb was stored in PBS and then subjected to uCT analysis (Sky Scan 1276 from Bruker, International) [25]. Measurements were made from end-to-end on femurs using MicroView software (GE Healthcare, Chicago, IL, USA) as previously described [71]. ANOVA statistic was used to compare the femoral measurements from mice limbs.

## Figures and Tables

**Figure 1 ijms-24-03845-f001:**
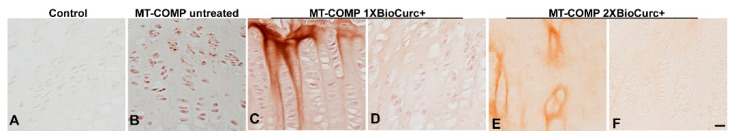
CurQ+ treatment reduces mutant-COMP protein ER retention. Control (C57BL\6) (**A**), MT-COMP (**B**), and MT-COMP growth plates treated with 1X (**C**,**D**), and 2X (**E**,**F**) CurQ+ from P7-P28 and were stained with human-COMP antibodies at P28 (**A**–**F**). Human-COMP antibody specifically recognizes human mutant-COMP expressed in response to DOX. Controls show no mutant-COMP protein staining (**A**) while untreated MT-COMP growth plate chondrocytes have intracellular retention (**B**), CurQ+ treatment allows some export of mutant-COMP protein (**C**,**E**), and these chondrocytes have very little mutant-COMP protein signal (**D**,**F**). Representative growth plates are shown from the examination of at least eight mice. Bar = 50 μm.

**Figure 2 ijms-24-03845-f002:**
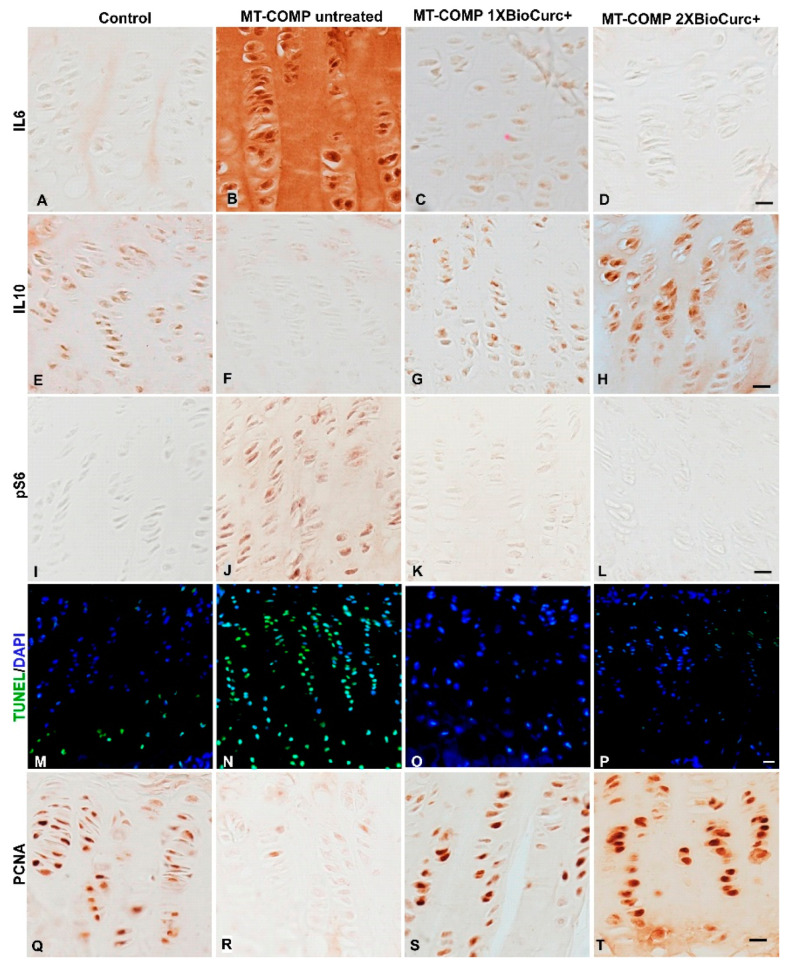
CurQ+ treatment normalizes MT-COMP growth plate chondrocytes. Growth plates at P28 from control (C57BL\6), MT-COMP, and MT-COMP mice treated with 1X and 2X CurQ+ doses from P7-P28 stained with interleukin 6 (IL6), interleukin 10 (IL10), phosphoS6 (pS6), and proliferating cell nuclear antigen (PCNA) antibodies and TUNEL (terminal deoxynucleotidyl transferase dUTP nick-end labeling) staining. The signal from inflammatory marker IL6 was minimal in controls and MT-COMP 1X and 2X CurQ+ treated growth plate chondrocytes compared to MT-COMP untreated controls (**A**–**D**). IL10 was abundant in controls (**E**) and in CurQ+ treated growth plates (**G**,**H**) and, absent in the MT-COMP growth plate (**F**). Autophagy was repressed in MT-COMP growth plates as shown by elevated pS6 immunostaining (measure of mTORC1 signaling) compared to controls and MT-COMP treated with CurQ+ (**I**–**L**). TUNEL-stained (green); DAPI shows nuclei (blue), and merged images are shown in (**M**–**P**). TUNEL-positive chondrocytes were found throughout the untreated MT-COMP growth plate, while there were, as expected, few TUNEL positive chondrocytes present only in the hypertrophic zone of control and MT-COMP CurQ+ treated growth plates. Chondrocyte proliferation was reduced in MT-COMP compared to controls and MT-COMP treated with CurQ+ (**Q**–**T**). Representative growth plates are shown from at least eight mice. Bar = 50 µm.

**Figure 3 ijms-24-03845-f003:**
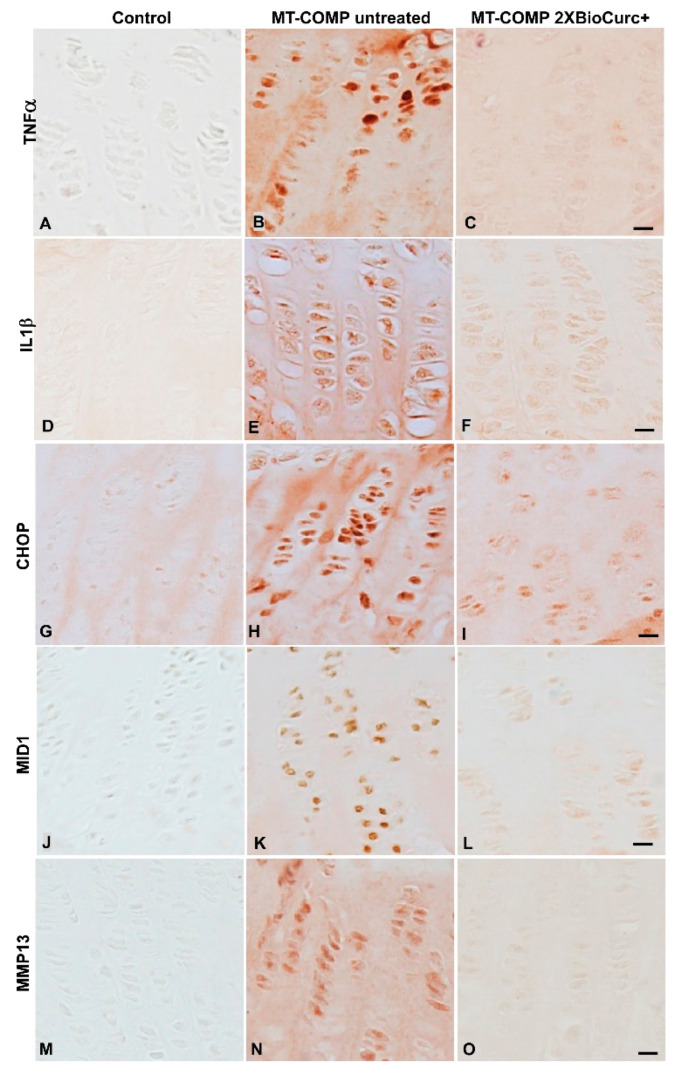
CurQ+ (2X) treatment normalized MT-COMP growth plate chondrocyte inflammation, anti-inflammatory, and ER stress markers. Growth plates at P28 from control (C57BL\6), MT-COMP, and MT-COMP mice treated with 2X CurQ+ from P7-P28 stained with tumor necrosis factor alpha (TNFα), interleukin 1β (IL1β), C/EBP homologous protein (CHOP), midline-1 (MID1), and matrix metalloprotease 13 (MMP13) antibodies. The signal from TNFα and IL1β, important inflammatory markers, are low in controls and MT-COMP 2X CurQ+ treated growth plate chondrocytes compared to MT-COMP untreated mice (**A**–**F**). The ER stress marker, CHOP, was reduced by 2X CurQ+ treatment but not eliminated (**G**–**I**). IL10 was abundant in controls (**E**) and in CurQ+-treated growth plates (**G**,**H**), and absent in the MT-COMP growth plate (**F**). MID1 and MMP13 signal was absent from controls and only scant staining was observed in 2X CurQ+ treated MT-COMP growth plates compared to untreated (**J**–**O**). Representative growth plates are shown from at least eight mice. Bar = 50 µm.

**Figure 4 ijms-24-03845-f004:**
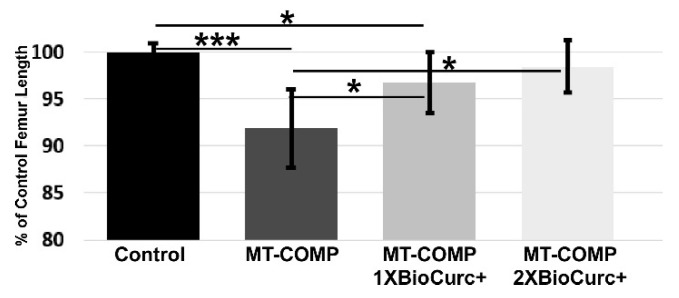
CurQ+ treatment increases MT-COMP murine femoral length. Femur lengths from CurQ+ treated with 1X and 2X doses from P7–P28 and untreated MT-COMP and control mice were measured by uCT at P28. At least 10 mice in each group were measured. Femur lengths were compared using an ANOVA test. P = postnatal day. * = *p* < 0.05; *** = *p* < 0.0005.

## Data Availability

Data is available upon request by writing to karen.posey@uth.tmc.edu within the limits of our patent application.

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
