# Peer review of "CurQ+, a Next-Generation Formulation of Curcumin, Ameliorates Growth Plate Chondrocyte Stress and Increases Limb Growth in a Mouse Model of Pseudoachondroplasia"

_ijms, 2023, doi:10.3390/ijms24043845_

Round 1

Reviewer 1 Report

    This manuscript reports on the use of a curcumin formulation, designated CurcQ+, to promote bone growth in a model of pseudoachondroplasia (PSACH). This model expresses the most common mutation found in cartilage oligomer matrix protein (COMP) in individuals with PSACH. The authors establish that CurcQ+ reduces the accumulation of mutant COMP in chondrocytes, and markedly reduces chondrocyte apoptosis and increases cellular proliferation. Importantly, CurcQ+ treatment results in a dose-dependent increase in bone length. These data are well-documented and clearly presented. The manuscript would be improved if the following points were addressed.

1.     The authors could further discuss the fact that mutant COMP is being secreted. This is not consistent with their model that posits that up-regulation of autophagy mediates the clearance of mutant COMP.

2.     Is there any precedent for “different secretory factors or ER quality control mechanisms” in the resting zone as compared to the rest of the growth plate?

3.     COMP is involved in multiple biological processes including blood pressure, pulmonary hypertension and cancer. Are the effects of mutant COMP limited to cartilage in PSACH patients? Might CurcQ+ treatment affect other aspects of PSACH?

4.     On page 7, the word plate is spelled platee.

Author Response

Notes:

Reviewers’ comments are in bold. 

Responses are in plain font indented. 

Text added to the manuscript are underlined and indented.  

Reviewer 1

  1. The authors could further discuss the fact that mutant COMP is being secreted. This is not consistent with their model that posits that up-regulation of autophagy mediates the clearance of mutant COMP.

In addition to adding text, a supplementary figure was added to show that the amount of COMP expressed plays a role in secretion.

Curcumin has been shown to induce autophagy1 and curcumin decreases pS6 (mTORC1 signaling marker) indicating that curcumin allows autophagy. The reviewer is correct that mutant-COMP protein export is contradictory to autophagy clearance hypothesis and the normal functions of protein quality control mechanisms.  In the manuscript, we state that and propose several processes that may allow export of mutant-COMP including 1) low level of expression in the resting zone of MT-COMP growth plate chondrocytes; 2) resting chondrocytes requiring some ER stress for differentiation; 3) potential differences in secretory/ER quality control factors in different zones.  This information is included in the manuscript in the following text. 

Previously, we have shown that turmeric treatments permit a small amount of extracellular export of mutant-COMP [27]; curcumin is the active ingredient in turmeric.  Interestingly, with CurcQ+ treatment mutant-COMP was observed in the extracellular matrix surrounding the resting zone chondrocytes, suggesting that ER quality control mechanisms that retain misfolded protein were either bypassed or overridden.  CurcQ+ stimulated export was limited to the resting zone and was not observed in the proliferating and hypertrophic zone of the growth plate at 4 weeks.  Our previous resveratrol work demonstrated that autophagy clears mutant-COMP protein [20, 22, 26, 27, 35] and this is likely the mechanism by which mutant-COMP protein is eliminated from the proliferating and hypertrophic chondrocytes with CurcQ+ treatment.  Curcumin has been shown to induce autophagy protecting cellular viability [66] and curcumin decreases pS6 (mTORC1 signaling marker) indicating that curcumin influences autophagy.  Interestingly, 1X CurcQ+ treated resting zone growth plate chondrocytes had more mutant-COMP protein in the extracellular matrix than the 2X dose.  Perhaps autophagy levels are higher in 2X CurcQ+ which reduces mutant-COMP export.  It is not known how some mutant-COMP escapes ER quality control systems and autophagy, but perhaps the lower levels of protein synthesis in the resting zone may not strongly trigger ER clearance mechanisms allowing the secretion of mutant-COMP.  Since professional secretory cells, such as chondrocytes, have basal ER stress that is essential for differentiation [67], and perhaps resting zone differentiation ER stress may allow export while not in chondrocytes in other zones are differentiated.  Alternatively, there could be different secretory factors or ER quality control mechanisms in the rest-ing zone that supports export that is not present in other growth plate zones.  This is supported by the observation that there is differential expression of ER stress related mRNAs driven by ER stress in the zones of the growth plate in another dwarfing con-dition, Schmid metaphyseal chondrodysplasia (MCDS), [68].

  1. Is there any precedent for “different secretory factors or ER quality control mechanisms” in the resting zone as compared to the rest of the growth plate?

There is evidence that the ER stress response is not equivalent in all zones in the growth plate and a statement has been added to the manuscript.

This is supported by the observation that there is differential expression of ER stress related mRNAs driven by ER stress in the zones of the growth plate in another dwarfing condition, Schmid metaphyseal chondrodysplasia (MCDS) [68].

  1. COMP is involved in multiple biological processes including blood pressure, pulmonary hypertension and cancer. Are the effects of mutant COMP limited to cartilage in PSACH patients? Might CurcQ+ treatment affect other aspects of PSACH?

There is no clinical evidence of cardiac abnormalities in PSACH individuals. However, curcQ+ may also prevent early joint degeneration that compromises the quality of life in PSACH.

  1. On page 7, the word plate is spelled platee.

This typographical error has been corrected.  Thank you for catching that.

Reviewer 2 Report

The authors described the effects of CurecQ+ against the pseudoachondroplasia induced by the expression of MT-COMP. These results are interesting enough and keep enough potential in prospects. However, results are only shown by immunohistochemical staining, without analyzing any molecular mechanism of this phenomenon, indicating that we can’t confirm the effects of this compound anymore.   

1. Authors described that autophagy induced by CurecQ+ decreased the ER stress and repressed the autophagy from the PERK pathway. Although this hypothesis is correct, the authors would indicate the activation of PERK in chondrocytes. First, the authors indicate that some amounts of MT-COMP are secreted as normal COMP by treatments of CurecQ+, why this MT-COMP escapes from the ER quality control system? It might be possible that the low amounts of MT-COMP do not induce the ER-stress and maturated as normal protein, but authors should show these possibilities by changing the amounts of DOX by the cell assay. Additionally, the amounts of secreted MT-COMP and dose of CurecQ+ were not proportional. The intensities of CurecQ+-induced autophagy might be different among the dose of CurecQ+, but it must be indicated by another assay system. Second, authors must show the induction of autophagy other than pS6 phosphorylation such as the appearance of LC3-II. Third, since the excess ER-stress induces the ER-phagy through activation of GSK3β, the authors did not check the ER-phagy still indicating the excess ER-stress induced apoptosis. Is there any contribution of the Rubicon, which causes the tissue-specific repression of autophagy?

2, Is it possible that activations of other ER-stress response pathways, such as the ATF6 pathway or the IRE1-XBP1pathway, are possible to repress the MT-COMP-induced autophagy? Or are they already activated by expression of the MT-COMP?

3, What kinds of cells produced several cytokines described in this manuscript? Infiltrated immune cells or chondrocytes themselves. Without distinguishing the kinds of cells, authors can’t indicate that CurecQ+ repressed PSACH through activation of autophagy of chondrocyte itself or inactivation of immune cells by suppressing the natural immunity of these cells.

Because of the above deficiencies, I don't think this manuscript is worth publication in IJMS.

Author Response

Reviewer 2   

1, Authors described that autophagy induced by CurQ+ decreased the ER stress and repressed the autophagy from the PERK pathway. Although this hypothesis is correct, the authors would indicate the activation of PERK in chondrocytes.

PERK is activated in chondrocytes that express mutant-COMP protein.  PERK activation and other details of ER stress evaluation have been included to address this concern.  

Previously, we demonstrated that all three ER stress sensors (PERK, ATF6, IRE1-XBP1) were activated by mutant-COMP expression early in the pathologic process but only the PERK pathway proceeded beyond activation leading to increased CHOP expression [21, 34].  Evaluation of Chop, Xbp1, Gadd45a, Gadd34, Ero1b, Grp78, and ER degradation enhancing alpha-mannosidase like protein 1 mRNA levels of (Edem1) [21] showed that Edem1 mRNA level was not changed in MT-COMP mice suggesting that endoplasmic-reticulum-associated protein degradation (ERAD) did not play a substantial role in the mutant-COMP pathology.  Importantly, we demonstrated that ER stress (CHOP), fundamental to the mutant-COMP protein pathology, is dramatically reduced by CurcQ+ treatment (Fig. 3G-I), thereby decreasing MID1 levels (Fig. 3J-L) that dampens a process that blocks autophagy preventing mutant-COMP protein clearance.  IL10, an anti-inflammatory protein, has been shown to reduce MMP13, an enzyme that degrades cartilage [49].  CurcQ+ 2X treatment increased IL10 levels (Fig. 2H) and consistent with its known action, MMP13 levels were reduced (Fig. 3M-O).

First, the authors indicate that some amounts of MT-COMP are secreted as normal COMP by treatments of CurQ+, why this MT-COMP escapes from the ER quality control system? It might be possible that the low amounts of MT-COMP do not induce the ER-stress and maturated as normal protein, but authors should show these possibilities by changing the amounts of DOX by the cell assay. Additionally, the amounts of secreted MT-COMP and dose of CurQ+ were not proportional. The intensities of CurQ+-induced autophagy might be different among the dose of CurQ+, but it must be indicated by another assay system.

This concern was addressed above see Reviewer 1 concern 1.  Importantly, we have shown that DOX dosage does not significantly alter the amount of mutant-COMP expressed2 and therefore evaluating LC3-II assay in cells will not resolve this issue.

Second, authors must show the induction of autophagy other than pS6 phosphorylation such as the appearance of LC3-II.

Previously, we have validated autophagy comparing the number LC3-II foci in chondrocytes in culture.  Text has been added to the manuscript describing these studies.  

Specifically, we have shown that mutant-COMP accumulation generates multiple cellular stresses in the MT-COMP growth plate chondrocytes, including ER stress (CHOP), oxidative stress, inflammation (TNFα), ECM degradative enzymes (MMP-13), block of autophagy (pS6; LC3 positive vesicles) [20, 35].  ER stress and tumor necrosis factor-alpha (TNFα) together increased midline 1 (MID1), a direct stimulator of mTORC1 signaling [20, 26, 35].  High levels of mTORC1 signaling favor protein expression over macroautophagy (hereafter referred to as autophagy).  Autophagy blockage drives continuous accumulation of mutant-COMP and other ECM proteins in the ER of chondrocytes, building an intracellular matrix that is not degraded by the cellular machinery, eventually overwhelming the cellular stress coping mechanisms and compromising viability [20, 26, 35].  Chondrocyte death is the result of oxidative stress that induces DNA damage and, ultimately, caspase-independent necroptosis [21, 34].  Importantly, these stresses were not observed in MT-COMP mice in the absence of DOX.

Third, since the excess ER-stress induces the ER-phagy through activation of GSK3β, the authors did not check the ER-phagy still indicating the excess ER-stress induced apoptosis.

GSK3 activates caspaces which induce cell death.  In previous work, we have shown that mutant-COMP induced chondrocyte death is caspase independent but rather is accomplished by necroptosis.  No ER-phagy receptors are changed in transcriptomic analysis

Chondrocyte death is the result of oxidative stress that induces DNA damage and, ul-timately, caspase-independent necroptosis [21, 34]. 

Is there any contribution of the Rubicon, which causes the tissue-specific repression of autophagy?

There is no evidence that rubicon plays a role in repression of autophagy in the context of MT-COMP mice since there is no difference in expression between mutant and wild-type (transcriptomic study E15, P0, P7, P14, P21, P28).  Our prior work shows that mTORC1 (AKT, mTOR, pS6) signaling is elevated by TNFα and ER stress and this was demonstrated in cultured cells treated with TNFα and ER stress inducing drugs (tunicamycin; thapsigargin)3.

2, Is it possible that activations of other ER-stress response pathways, such as the ATF6 pathway or the IRE1-XBP1pathway, are possible to repress the MT-COMP-induced autophagy? Or are they already activated by expression of the MT-COMP?

All UPR sensors have been evaluated previously and text has been added to the manuscript for clarification.

Previously, we demonstrated that all three ER stress sensors (PERK, ATF6, IRE1-XBP1) were activated by mutant-COMP expression early in the pathologic process but only the PERK pathway proceeded beyond activation leading to increased CHOP expression [21, 34].  Evaluation of Chop, Xbp1, Gadd45a, Gadd34, Ero1b, Grp78, and ER degradation enhancing alpha-mannosidase like protein 1 mRNA levels of (Edem1) [21] showed that Edem1 mRNA level was not changed in MT-COMP mice suggesting that endoplasmic-reticulum-associated protein degradation (ERAD) did not play a substantial role in the mutant-COMP pathology.

3, What kinds of cells produced several cytokines described in this manuscript? Infiltrated immune cells or chondrocytes themselves. Without distinguishing the kinds of cells, authors can’t indicate that CurQ+ repressed PSACH through activation of autophagy of chondrocyte itself or inactivation of immune cells by suppressing the natural immunity of these cells.

Chondrocytes in both the growth plate and articular cartilage express pro-inflammatory proteins and a statement has been added to the manuscript text.   

Growth plate chondrocytes are known to produce pro-inflammatory cytokines [44]; we have shown that mutant-COMP expression increased expression of multiple inflam-matory proteins [20, 22, 26, 27, 35].

Because of the above deficiencies, I don't think this manuscript is worth publication in IJMS.

Much of Reviewer 2’s comments revolve around evidence of ER stress and autophagy.  We have done detailed studies of both of these processes and have included this information in the manuscript for clarity.  Importantly, this manuscript builds on our prior work demonstrating the efficacy of CurcQ+ to repress the mutant-COMP chondrocyte phenotype.  This manuscript is not focused on delineated the mechanisms that are involved in mutant-COMP chondrocyte pathology as we have published work describing these mechanisms. 

[1] Forouzanfar F, Read MI, Barreto GE, Sahebkar A: Neuroprotective effects of curcumin through autophagy modulation. IUBMB Life 2020, 72:652-64.

[2] Posey KL, Veerisetty AC, Liu P, Wang HR, Poindexter BJ, Bick R, Alcorn JL, Hecht JT: An Inducible Cartilage Oligomeric Matrix Protein Mouse Model Recapitulates Human Pseudoachondroplasia Phenotype. The American Journal of Pathology 2009, 175:1555-63.

[3] Posey KL, Coustry F, Veerisetty AC, Hossain MG, Gambello MJ, Hecht JT: Novel mTORC1 Mechanism Suggests Therapeutic Targets for COMPopathies. Am J Pathol 2019, 189:132-46.

Round 2

Reviewer 2 Report

The authors described the effects of CurQ+ against the pseudoachondroplasia induced by the expression of MT-COMP. These results are interesting enough and keep enough potential in prospects. However, results are only shown by immunohistochemical staining, without analyzing any quantification and appropriate control experiments. I point out some defects of this manuscript, but these are not all.

1. In Figure 1 to 3, the authors can’t describe anything without appropriate quantification analysis.

2. However mutant-COMP transported via ER, the localization of induced mutant-COMP in cells is not determined at ER without any double staining of ER marker molecules. It might be possible that induced mutant-COMP is accumulated in cytosol after retro translocation out of ER or stored in lysosome before degradation.

3. Are the mutant-COMP in ECM are matured form? These are simply examined by checking glycosidation of these proteins. By checking these conditions, the localizations of mutant-COMP under experimental conditions would be investigated more specifically.

4. The authors should show the translation rate of the mutant-COMP is not changed after addition of CurQ+, because it might be possible that inhibitions of translation of the mutant-COMP reduced ER stress and resulted in appropriate maturation environments for mutant-COMP.

5. It is well known that CurQ+ shows reduction effect and these effects induces the ER stress responses. Without more detailed experiments, the authors can never conclude that the autophagy is responsible for these experiments.

6. There are several deficiency for abbreviations, such as P7-P28, OA. Additionally, there are a little miss spelling (TNFα; IL1β).

Because of the above deficiencies, I don't think this manuscript is worth publication in IJMS.